# Increase in Double Negative B Lymphocytes in Patients with Systemic Lupus Erythematosus in Remission and Their Correlation with Early Differentiated T Lymphocyte Subpopulations

Eleni Moysidou [1,2], Georgios Lioulios [1,2], Michalis Christodoulou [1,2], Aliki Xochelli [3], Stamatia Stai [1,2], Myrto Iosifidou [1], Artemis Iosifidou [1], Sophia Briza [4], Dimitria Ioanna Briza [5], Asimina Fylaktou [3] and Maria Stangou [1,2,*]

1   School of Medicine, Aristotle University of Thessaloniki, 45636 Thessaloniki, Greece; moysidoueleni@yahoo.com (E.M.); pter43@yahoo.gr (G.L.); michalischristodoulou22@gmail.com (M.C.); staimatina@yahoo.gr (S.S.); ios.myrtv2002@gmail.com (M.I.); ios.artemis.2000@gmail.com (A.I.)
2   Department of Nephrology, General Hospital "Hippokration", 54642 Thessaloniki, Greece
3   Department of Immunology, National Histocompatibility Center, General Hospital "Hippokration", 54642 Thessaloniki, Greece; aliki.xochelli@gmail.com (A.X.); fylaktoumina@gmail.com (A.F.)
4   Department of Architecture, School of Engineering, University of Thessaly, 38334 Thessaly, Greece; s_briza@yahoo.com
5   School of Informatics, Aristotle University of Thessaloniki, 45636 Thessaloniki, Greece; ddmpriza@gapps.auth.gr
*   Correspondence: mstangou@auth.gr; Tel.: +30-2310992788

**Abstract:** B and T lymphocytes demonstrate important alterations in patients with systemic lupus erythematous (SLE), with a significant upregulation of double negative (DN) B cells. The aim of this study was to evaluate the correlation of T cell immunity changes with the distinct B-cell-pattern SLE. In the present study, flow cytometry was performed in 30 patients in remission of SLE and 31 healthy controls to detect DN B cells (CD19+IgD-CD27-) and a wide range of T lymphocyte subpopulations based on the presence of CD45RA, CCR7, CD31, CD28, and CD57, defined as naïve, memory, and advanced differentiated/senescent T cells. Both B and T lymphocytes were significantly reduced in SLE patients. However, the percentage of DN B cells were increased compared to HC (12.9 (2.3–74.2) vs. 8 (1.7–35), $p = 0.04$). The distribution of CD4 and CD8 lymphocytes demonstrated a shift to advanced differentiated subsets. The population of DN B cells had a significant positive correlation with most of the early differentiated T lymphocytes, CD4CD31+, CD4CD45RA+CD28+, CD4CD45RA+CD57-, CD4CD45RA-CD57-, CD4CD28+CD57-, CD4CD28+CD57+, CD4 CM, CD8 CD31+, CD8 NAÏVE, CD8CD45RA-CD57-, CD8CD28+CD57-, and CD8CD28+CD57+. Multiple regression analysis revealed CD4CD31+, CD8CD45RA-CD57-, and CD8CD28+CD57- cells as independent parameters contributing to DN B cells, with adjusted $R^2 = 0.534$ and $p < 0.0001$. The predominance of DN B cells in patients with SLE is closely associated with early differentiated T lymphocyte subsets, indicating a potential causality role of DN B cells in T lymphocyte activation.

**Keywords:** systemic lupus erythematosus; lupus nephritis; double negative B lymphocytes; T lymphocytes; early differentiated cells; senescent lymphocytes

## 1. Introduction

B lymphocytes represent about 10–20% of the total circulating lymphocytes. They can be divided into four major subpopulations based on the presence of IgD and CD27 surface molecules [1]. IgD, which is expressed by naïve B cells, progressively diminishes and is eliminated after isotype switching. Instead, CD27, a receptor molecule belonging to the tumor necrosis factor (TNF) superfamily, is not expressed by naïve B lymphocytes, yet it appears at later differentiation stages, and its presence signifies memory B cells [2,3]. Therefore, IgD+CD27- cells are described as naïve, IgD+CD27+ cells as non-switched

memory, and IgD-CD27+ as switched memory B cells. Although the above three subtypes are clearly defined in terms of origin, differentiation, and function, the fourth one, double negative (DN) B cells, namely IgD-CD27-, is still an enigmatic subpopulation. The absence of IgD indicates isotype switching, while the absence of CD27 specifies fewer differentiated cells [4,5]. For many years, these cells were characterized as a naïve B cell population, and they were under-recognized; yet, only recently, their origin has caused major interest. Hypothetical models of their derivation have been described based on their phenotypic and functional characteristics. Therefore, based on the increased expression of B cell lymphoma protein, similar to that of CD27+ cells, DN B cells are anticipated to originate after defective reactions in germinal centers or to derive from memory B cells following the displacement or downregulation of the CD27 molecule during the immune-senescent or immune-exhaustion process [6–8].

DN B cells, described as DN1 and DN2, according to the presence or absence of CD24, CD38, and CXCR5 proteins, respectively, are increased during aging and also in chronic inflammatory or systemic diseases as a result of chronic stimulation [4]. Systemic lupus erythematosus is characterized by the increased expression of DN2 B cells, probably as a result of a hyperresponsiveness to Toll-like receptor-7 signaling [9]. Recent studies have shown that DN2 B cells were mainly increased in active SLE and in the presence of lupus nephritis, and they were positively correlated with anti-RNA, anti-Sm, and anti-RNP autoantibodies [9–11]. We have previously described the predominance of DN B cells in SLE patients, even compared to patients on hemodialysis, selected as a representative group of patients with senescent adaptive immunity [11–13].

Based on the previously described significant increase in DN B lymphocytes in SLE patients and their important role in autoimmunity, we contacted the present study to evaluate their possible correlation with T lymphocyte alterations, also common in SLE patients. T lymphocytes were divided into subpopulations according to their differential status, and the potential correlation with DN B cells was estimated.

## 2. Materials and Methods

After analyzing the phenotype of peripheral B and T lymphocytes, we estimated the frequencies of DN B lymphocytes and assessed the potential correlation between them and T lymphocyte subsets.

### 2.1. Study Design and Population

This is a cross-sectional study. We included patients in remission of SLE and healthy controls. The study population involved Caucasian SLE patients and healthy controls (HC) with similar age, sex, and ethnicity. The diagnosis of SLE was based on the SLICC/ACR 2012 criteria [14]. The diagnosis of lupus nephritis was based on kidney biopsy findings.

Inclusion criteria: All participants were adults aged 18–67 years. Patients had to be under regular follow-up with available clinical, biochemical, and immunological data. SLE patients had to be stable with no flare-ups for at least 2 years, and their immunosuppression at the time of evaluation could include prednisolone, hydroxychloroquine, calcineurin inhibitors, azathioprine, or mycophenolate mofetil.

Exclusion criteria: Comorbidities such as diabetes mellitus; malignancy; hematological disorder; impaired renal function, defined as eGFR < 60 mL/min/m$^2$; or the presence of chronic active or recent (<6 months) infection. Additionally, patients who had received monoclonal antibodies in the past or cyclophosphamide during the last 24 months were also excluded.

Inclusion–Exclusion criteria for the HC group: Volunteers eligible to participate in the HC group had to be Caucasians, 18–67 years old, with no comorbidities, as described above, and with normal renal function. They could not have been taking any medication, and caution was taken to be of similar sex with patients. HC were either patients' partners or medical or paramedical personnel.

Information regarding previous historical data, medication, comorbid conditions, and disease flares was retrieved from patients' records. At the time of assessment, patients' biochemical and immune profiles were recorded, and the SLEDAI score was used to estimate the activity of SLE [15].

The study was conducted in accordance with the Declaration of Helsinki and approved by the Institutional Review Board of the Medical School of Aristotle University of Thessaloniki, Greece, protocol code 748/21, October 3 2021. All participants were informed prior to enrollment and signed a written consent. Additionally, written informed consent has been obtained from the patients to publish this paper.

### 2.2. Flow Cytometry

Heparinized venous blood from SLE patients and healthy control individuals was drowned under sterile conditions and collected in EDTA tubes. Within 12 h following collection, samples proceeded for the evaluation of total white cell count, total lymphocytes, B and T lymphocytes, and further analysis for CD4 and CD8 subtypes.

The expression of IgD and CD27 on B lymphocytes and of CD45RA, CCR7, CD28, CD31, and CD57 on both CD4 and CD8 T lymphocytes was assessed using a cytometer (Navios Flow Cytometer, Beckman Coulter, Brea, CA, USA) as described before [8]. The expression of the above surface molecules examined determined divergent subpopulations. The proportions and counts of all lymphocyte subsets were estimated.

The monoclonal antibodies used in flow cytometry to identify lymphocyte surface receptors were anti-CD45 PC7 J33 (IM3548U, Beckman Coulter), anti-CD19 PC5 J3-119 (Beckman, Coulter), anti-IgD IA6-2 (ThermoScientific LSG), and anti-CD27 PE-DYLIGH 594 (EXBIO, Praha SA) for B lymphocytes, and anti-CD45 PC7 J33 (IM3548U, Beckman Coulter), anti-CD3 FITC UCTH1 (A07746, Beckman Coulter), anti-CD4 Pacific Blue MEM-241 (PB-359-T100, EXBIO, Praha SA), anti-CD8 PC5 B9.11(A7758, Bechman Coulter), anti-CD45RA APC MEM-56 (1A-223-T100, EXBIO, Praha SA), anti-CCR7 PE 4B12 (1P-735-C100,EXBIO Praha SA), anti-CD28 CD28.2 PE-EF610 (61-0289-42, ThermoScientific LSG), anti-CD57 FITC TB01(1F-158-T100, EXBIO), and CD31 ECD for CD4 and CD8 lymphocytes.

We used three different tubes (for B, CD4, and CD8 lymphocytes), and a combination of estimated receptors resulted in the definition of the following subpopulations.

### 2.3. Definition of Lymphocyte Subpopulations

B lymphocyte subpopulation definition was based on the expression of CD27 and IgD surface molecules and described as naïve (CD19+IgD+CD27-), non-switched and switched memory (CD19+IgD+CD27+ and CD19+IgD+CD27-, respectively), and double negative (DN) B cells (CD19+IgD-CD27-).

The classification of CD4 and CD lymphocytes was based on the presence of the above-described surface receptors and described as early differentiated, memory, and senescent/advanced differentiated cells.

1. Early differentiated CD4 or CD8 lymphocytes that expressed:
    1. CD31, CD4+CD31+, and CD8+CD31+, defined as Recent Thymic Emigrants (RTE);
    2. CD45RA together with CCR7, CD4+CD45RA+CCR7+, and CD8+CD45RA+CCR7+, defined as naïve CD4 or CD8 lymphocytes;
    3. CD28 and not CD57, CD4CD28+CD57-, and CD8CD28+CD57-;
    4. CD45RA and not CD57, CD4CD45RA+CD57-, and CD8CD45RA+CD57-;
    5. Neither CD45RA nor CD57, CD4CD45RA-CD57-, and CD8CD45RA-CD57-.

2. In the group of memory cells, CD4 or CD8 lymphocytes that expressed:
    1. CCR7 and not CD45RA, CD4+CD45RA-CCR7+, and CD8+CD45RA-CCR7+, are defined as central memory (CM);
    2. Neither CCR7 nor CD45RA, CD4+CD45RA-CCR7- and CD8+CD45RA-CCR7-, are defined as effector memory (EM).

3. In the group of senescent/advanced differentiated cells:

1.  T cells expressing CD45RA and not CCR7, CD4+CD45RA+CCR7-, and CD8+CD45RA +CCR7-, are defined as EMRA;
2.  T cells lacking the CD28 molecule, CD4CD28-, and CD8CD28-, and their subtypes according to the presence of CD57, CD4+CD28-CD57-, CD4+CD28-CD57+, and CD8+CD28-CD57-, CD8+CD28-CD57+;
3.  T cells expressing both CD45RA and CD57, CD4CD45RA+CD57+, and CD8CD 45RA+CD57+.

Gating strategies for B and T lymphocyte subpopulations are described in the Supplementary File, Figures S1 and S2.

*2.4. Statistical Analysis*

Statistical analysis was performed using the Statistical Package for Social Sciences (SPSS Inc., Chicago, IL, USA) version 27.0 for Windows. All continuous variables were initially tested for normality of distribution by using the Shapiro–Wilk and/or the Kolmogorov–Smirnov test.

Normally distributed variables were expressed as mean ± standard deviation, and Student's *t* test for independent samples was performed to compare differences between two groups. Data from non-parametric variables were expressed as median and range, and Mann–Whitney U test was used to assess differences between two groups. Pearson's and Spearman's correlation tests were applied to compare variables with normal or non-normal distribution, respectively. Multiple regression analysis was performed to estimate independent factors correlated with DN B lymphocytes as the dependent variable.

Values of $p < 0.05$ (two-tailed) were considered statistically significant.

**3. Results**

*3.1. Characteristics of SLE Patients*

Thirty-seven patients were consecutively evaluated initially, from whom seven were excluded: three because of a recent (<2 years) relapse; two because they had rituximab in the past; one had a history of melanoma; and one had developed diabetes mellitus. All patients were followed up in the SLE outpatient clinic and recruited within 6 months.

Thirty patients with SLE (twenty-nine females) and 31 healthy individuals (17 females) of similar age, sex, and ethnicity were finally included in the study. Females predominated in the SLE patient group; however, all parameters tested did not show any difference between males and females, neither in the patients nor in the HC group.

According to the inclusion criteria, all SLE patients were in complete or partial remission and under stable treatment for the last 24 months. Their treatment included hydroxychloroquine in 30/30 (100%), prednisolone (5–7.5 mg/d) in 24/30 (80%), MMF (500–1000 mg/d) in 18/30 (60%), and CNIs (1.5 mg/Kg/d) in 9/30 (30%) or a combination of MMF+CNIs in 5/30 (16.7%). The time since patients were on the above regimens was 84 (45–125) months for hydroxychloroquine, 78 (45–92) months for steroids, 53 (37–65) months for MMF, and 42 (28–56) months for CNIs.

Twenty patients (67%) had been given cyclophosphamide in the past, more than 2 years ago, while no one had received monoclonal antibodies in the past.

Clinical and laboratory findings of patients are depicted in Table 1.

Median time since the initial diagnosis of SLE was 84 (45–125) months. SLEDAI score was estimated at 10 (2–18) at the time of diagnosis and 2 (1–5) at the time of evaluation. Twenty-two patients (73.3%) had been LN diagnosed with renal biopsy performed not more than 5 years prior to evaluation. Mean SCr at the time of assessment was 0.89 ± 0.17 mg/dL, eGFR 81 ± 19 mL/min/1.73 m$^2$, Uprot 680 ± 5.600 mg/24h, and C3 and C4 levels at the time of evaluation were 74.65 (29.9–127) mg/dL and 15.7 (3.86–27.2) mg/dL, respectively.

**Table 1.** Clinical and laboratory characteristics of SLE patients and healthy controls at the time of assessment.

|  | SLE | HC | *p* |
|---|---|---|---|
| n | 30 | 31 |  |
| Laboratory results |  |  |  |
| WCC (cells/µL) | 7200 ± 3350 | 6400 ± 1800 | NS |
| Neutrophils (%) | 69.6 ± 20.7 | 58.1 ± 10.35 | <0.0001 |
| Neutrophils (cells/µL) | 4600 ± 3500 | 3500 ± 1200 | 0.03 |
| Lymphocytes (%) | 23.1 ± 16.1 | 25.6 ± 9 | 0.03 |
| Lymphocytes (cells/µL) | 1400 ± 900 | 2100 ± 900 | 0.005 |
| NLR | 3 ± 4 | 1.8 ± 0.85 | <0.0001 |
| Ht (%) | 33 ± 9 | 39 ± 5 | 0.02 |
| Hb (mg/dL) | 11.5 ± 6 | 13 ± 5 | 0.03 |
| Platelets ($10^3$/µL) | 245 ± 96 | 270 ± 35 | NS |
| CRP | 0.5 ± 0.4 | 0.4 ± 0.6 | NS |
| Serum urea (mg/dL) | 36 ± 12 | 34 ± 5 | NS |
| Serum creatinine (mg/dL) | 0.9 ± 0.5 | 0.87 ± 6 | NS |
| Total protein (g/dL) | 7.5 ± 4 | 7.7 ± 3.2 | NS |
| Serum albumin (g/dL) | 4.2 ± 2.3 | 4.6 ± 2.5 | NS |

*3.2. Phenotypic Analysis of B Lymphocytes in Patients with SLE and in HC*

Differences in B lymphocytes and their subpopulations are shown in Table 2. SLE patients had a significantly reduced percentage of B cells and also reduced proportions of CD19+IgD+CD27+ and CD19+IgD-CD27+ cells, with a clear shift to CD19+IgD-CD27- (DN) B cells. The population of B lymphocytes was significantly reduced compared to controls, almost 1/3 of the HC group, and this reduction affected CD19+IgD+CD27- and CD19+IgD+CD27+ cells (Table 2).

**Table 2.** CD19+ lymphocytes and their subpopulations. Differences in the percentages and populations between SLE patients and HC.

|  | SLE | HC | *p* |
|---|---|---|---|
| n | 30 | 31 |  |
| CD19 (%) | 7.9 (2.1–28.6) | 11.8 (5.4–24) | 0.012 |
| CD19 cells/µL | 75.4 (14.4–520.8) | 214 (84–576) | <0.001 |
| IgD+CD27- (%) | 51.5 (0.4–94) | 58.7 (4.5–86.9) | 0.34 |
| IgD+CD27- cells/µL | 37.71 (0.26–434.84) | 117 (5–364) | <0.001 |
| IgD+CD27+ (%) | 3.9 (0.2–22) | 8.4 (1.5–44) | 0.014 |
| IgD+CD27+ cells/µL | 5.12 (0.13–17.55) | 23 (2–700) | <0.001 |
| IgD-CD27+ (%) | 19.1 (2.2–78) | 17.9 (7.1–71.9) | 0.7 |
| IgD-CD27+ cells/µL | 18.58 (0.47–89.58) | 38 (11–258) | 0.001 |
| IgD-CD27- (%) | 12.9 (2.3–74.2) | 8 (1.7–35) | 0.04 |
| IgD-CD27- cells/µL | 10.84 (0.93–122.91) | 21 (3–202) | 0.007 |
| Ratio DN/[ (IgD+CD27-) + (IgD-CD27+) + (IgD+CD27+)] | 0.14 (0.02–2.9) | 0.08 (0.02–0.54) | *p* = 0.04 |

As the DN B cell population was the only one with an increased percentage within B lymphocytes, we calculated the ratio of (IgD-CD27-)/[(IgD+CD27-) + (IgD-CD27+) + (IgD+CD27+)] in SLE patients and controls and found a significant difference of 0.14 (0.02–2.9) vs. 0.08 (0.02–0.54), respectively; *p* = 0.04.

The distribution of B lymphocyte subtypes and the differences between SLE patients and controls are depicted in Figure 1.

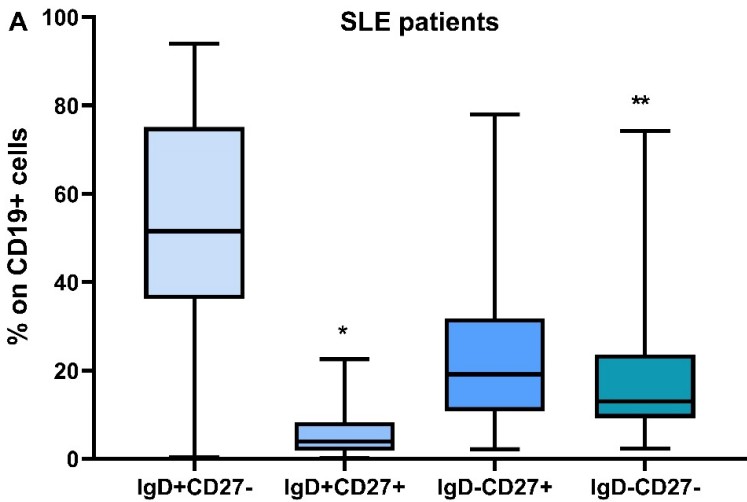

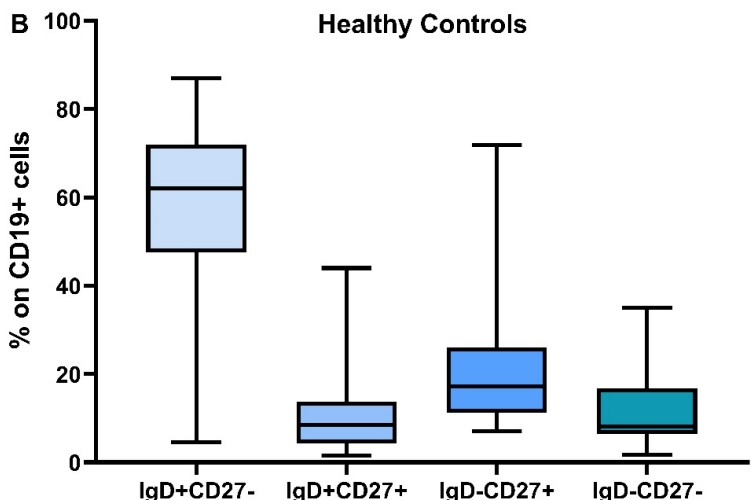

**Figure 1.** Distribution of B lymphocyte subtypes, IgD+CD27-, IgD+CD27+, IgD-CD27+, IgD-CD27-, in SLE patients (**A**) and healthy controls (**B**), * Difference between SLE patients and HC, *p* = 0.014, ** Difference between SLE patients and HC, *p* = 0.04.

### 3.3. Phenotypic Analysis of T Lymphocytes in SLE Patients and HC

CD4 and CD8 lymphocyte subpopulations were classified as early differentiated, memory, and advanced differentiated/senescent cells. Classification was based on their phenotypic characteristics, namely the presence of specific surface receptors.

In SLE patients, there was a significant reduction in the whole cohort of CD4 but not in the CD8 lymphocytes. More specifically, almost all early differentiated and memory CD4 subtypes were significantly reduced in SLE patients, as were the CD4 EMRA cells. The expression of the CD28 molecule on CD4 lymphocytes was reduced but not statistically significant (Table 3).

Regarding CD8 lymphocytes, CD8EMRA (CD8CD45RA+CCR7-) and CD8EMRACD28-(CD8CD45RA+CCR7-CD28-) cells showed a statistically significant reduction in SLE patients compared to controls, which proved a decrease in the senescent phenotype, together with a shift to CM and early differentiated CD8 subtypes (Table 4).

### 3.4. Distribution of CD4 and CD8 Subtypes According to Their Differentiation Status

In Figures 2 and 3, the differences in CD4 and CD8 subpopulations are depicted. According to the presence or absence of CD45RA, CD28, and CD57 molecules, T lymphocytes were divided into subgroups, and their percentages were calculated in SLE patients and HC.

**Table 3.** Differences in CD4 lymphocytes and their subpopulations between SLE patients and HC.

| | SLE | HC | *p* |
|---|---|---|---|
| n | 30 | 31 | |
| CD4 (cells/μL) | 651.2 (71.1–1478.2) | 986 (344–1591) | 0.004 |
| Early differentiated cells | | | |
| CD4+CD31+ | 216.38 (16.3–904.7) | 250 (69–967) | 0.14 |
| CD4CD45RA+CD28+ | 267.97 (20.62–1030.31) | 388 (139–1402) | 0.02 |
| CD4CD45RA+CD57- | 254.03 (21.05–1077.61) | 401 (160–1373) | 0.035 |
| CD4CD45RA-CD57- | 290.67 (38.96–884.43) | 539 (173–991) | <0.001 |
| CD4CD28+CD57- | 610.7 (54.68–1461.94) | 958 (332–1569) | 0.004 |
| CD4CD28+CD57+ | 4.7 (0–806) | 7 (0–245) | 0.21 |
| Memory cells | | | |
| CD4CD45RA-CCR7+ | 402.35 (38.7–972.4) | 563 (40–1001) | 0.046 |
| CD4CD45RA-CCR7- | 1.62 (0–73.49) | 11 (0–590) | 0.002 |
| Advanced differentiated/senescent cells | | | |
| CD45RA+CCR7- | 7.29 (0–180.62) | 23 (0–487) | 0.027 |
| CD4CD28- | 20.12 (1.27–139.06) | 38 (3–299) | 0.04 |
| CD4CD28-CD57+ | 9.90 (0.46–73.8) | 23 (0–274) | 0.1 |
| CD45RA+CCR7-CD28- | 1.2 (0–82) | 2.5 (0–106) | 0.21 |

**Table 4.** Differences in CD8 lymphocytes and their subpopulations between SLE patients and HC.

| | SLE | HC | *p* |
|---|---|---|---|
| n | 30 | 31 | |
| CD8 (cells/μL) | 414.8 (60.6–2017.8) | 454.5 (154–1310) | 0.26 |
| Early differentiated cells | | | |
| CD8+CD31+ | 88.19 (8.2–1047) | 187.5 (8–541) | 0.26 |
| CD8CD45RA+CD28+ | 113.56 (1.81–753.7) | 212.5 (7–1257) | 0.17 |
| CD8CD45RA+CD57- | 63.65 (3.83–889.8) | 133 (8–552) | 0.17 |
| CD8CD45RA-CD57- | 194.52 (1.8–945.1) | 179 (28–555) | 0.99 |
| CD8CD28+CD57- | 249.45 (5.49–1362) | 298 (95–646) | 0.1 |
| CD8CD28+CD57+ | 12(0.4–132) | 8.5 (0–424) | 0.58 |
| Memory cells | | | |
| CD8CD45RA-CCR7+ | 171.52 (2.5–1417) | 123 (1–941) | 0.14 |
| CD8CD45RA-CCR7- | 13.94 (0.59–92.37) | 25 (0–355) | 0.53 |
| Advanced differentiated/senescent cells | | | |
| CD8CD45RA+CCR7- | 11.13 (0–279.6) | 49.5 (0–534) | 0.02 |
| CD8CD28- | 87.83 (4.56–1361.2) | 135 (36–633) | 0.14 |
| CD8CD28-CD57+ | 53.17 (0.83–571.04) | 71 (0–470) | 0.17 |
| CD45RA+CCR7-CD28- | 37.3 (2.1–263) | 197 (9–783) | <0.0001 |

*3.5. Correlation of DN B Cells with B Lymphocyte Subpopulations*

Figure 4 demonstrates the correlation of DN B cells with the rest of the B cell subpopulations in SLE and HC. Multiple regression analysis with DN cells as dependent and the rest of B cell populations as independent factors showed that in both patients and controls, CD19IgD-CD27+ was the only independent parameter correlated to DN B cells; $R^2$ = 0.68, *p* = 0.001 for patients and $R^2$ = 0.7, *p* < 0.0001 for HC.

*3.6. Correlation of DN B Cells with T Lymphocyte Subpopulations*
3.6.1. Correlation with CD4 Cells

Both in patients and controls, DN B cells showed a significant positive correlation with the number of CD4 cells. In HC, DN B cells showed a positive correlation with a few of the early differentiated CD4 subsets, namely CD4CD45RA+CD57- and CD4CD28+CD57-, while in SLE patients, almost all early differentiated CD4 subtypes had a significant positive correlation with DN B cells, RTEs, and all subtypes lacking CD57 and expressing CD28

on their membrane. CM CD4 cells also correlated to DN B cells (Figure 5, Supplementary Table S2).

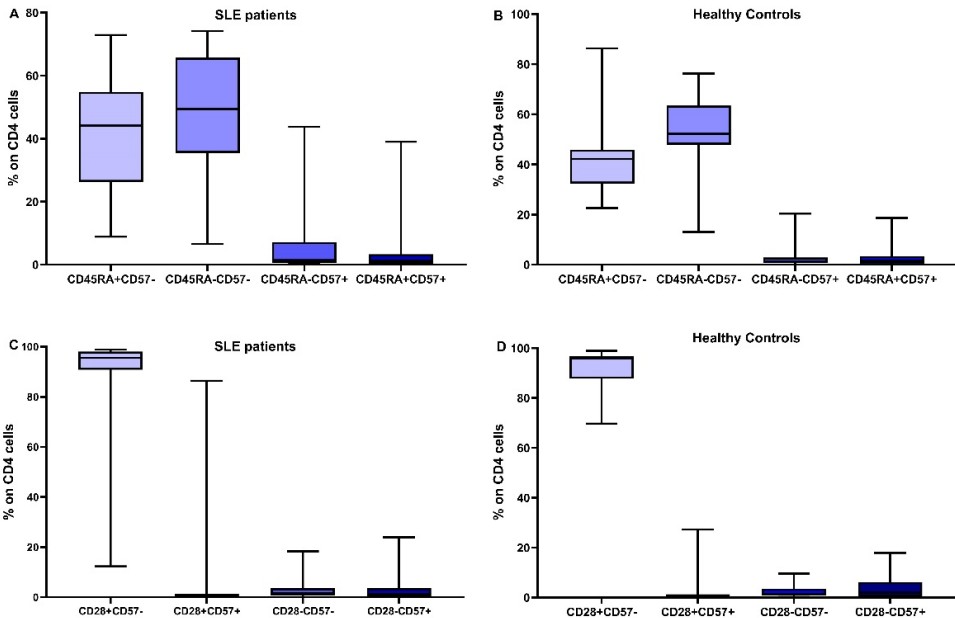

**Figure 2.** Distribution of CD4 subtypes: the combination of CD45RA expression and CD57 in SLE patients (**A**) and healthy controls (**B**), CD28 and CD57 in SLE patients (**C**) and healthy controls (**D**).

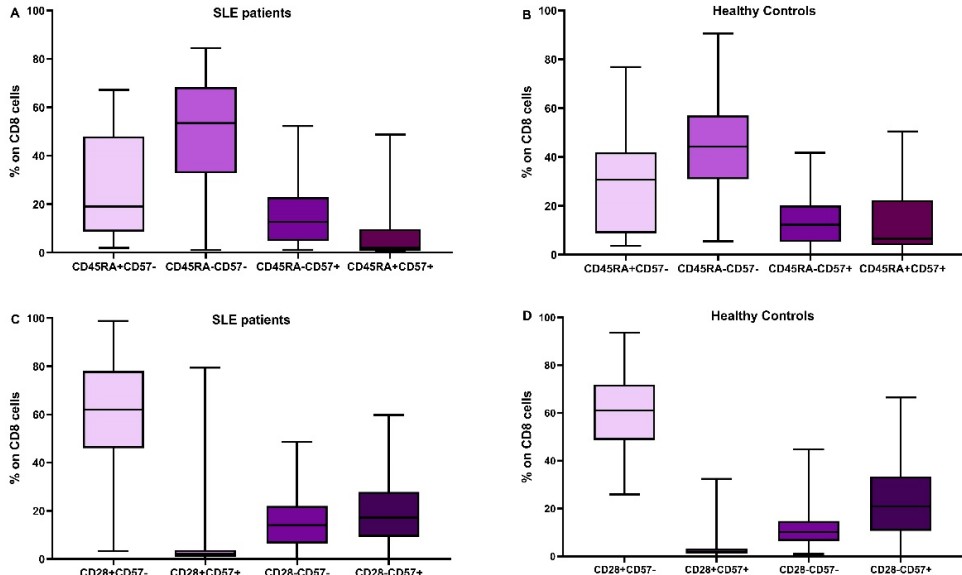

**Figure 3.** Distribution of CD8 subtypes: the combination of CD45RA expression and CD57 in SLE patients (**A**) and healthy controls (**B**), CD28 and CD57 in SLE patients (**C**) and healthy controls (**D**).

### 3.6.2. Correlation with CD8 Cells

Regarding CD8 lymphocytes, DN B cells in HC had no significant correlation, neither with CD8 cells nor with their subpopulations. On the other hand, in SLE patients, DN B cells showed a major association with RTEs, naïve CD8 cells, CD8CD45RA-CD57-, CD8CD28+CD57-, and CD8CD28+CD57+. No correlation was evident with memory or senescent CD8 cells (Figure 6, Supplementary Table S2).

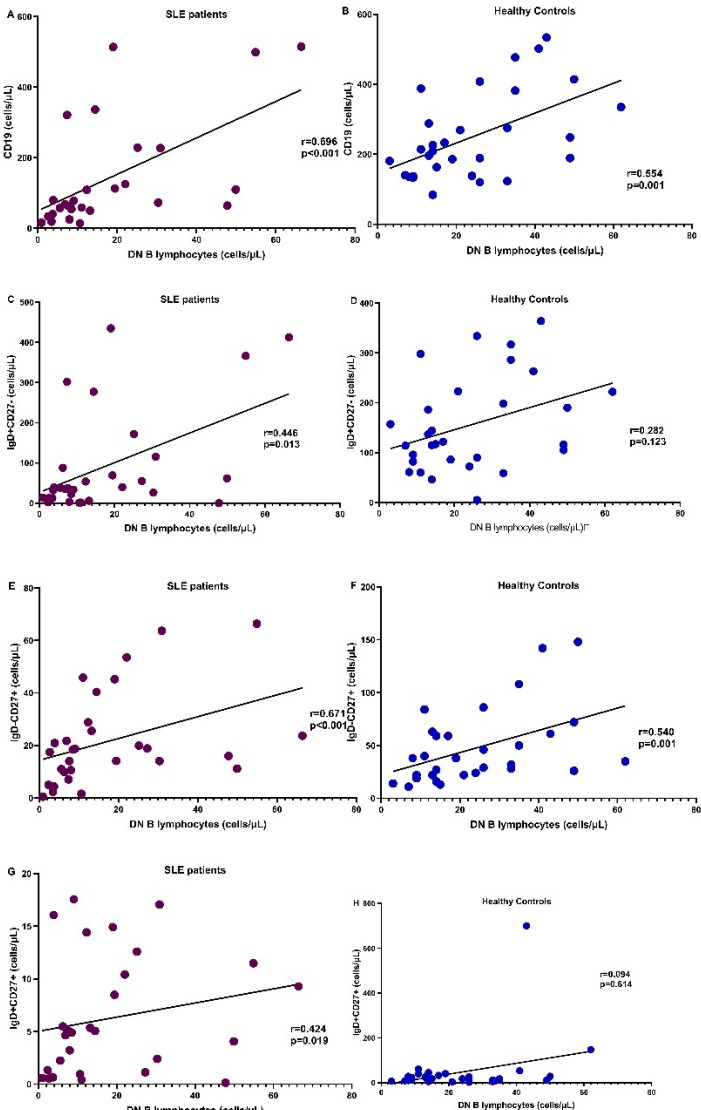

**Figure 4.** Correlation of the number of DN B cells with the whole cohort of CD19+ cells (**A**,**B**), CD19IgD+CD27- (naïve) (**C**,**D**), CD19IgD-CD27+ (non-switched) (**E**,**F**), and CD19IgD+CD27+ (switched memory) (**G**,**H**) B cells in SLE patients (purple) and HC (blue), respectively.

Correlations of DN B cells with CD4 and CD8 lymphocyte counts and their subsets are depicted in Figures 5 and 6, demonstrating the significant association with early differentiated T lymphocytes.

Multiple regression analysis, including DN B cells as a dependent variable and CD4 and CD8 as independent variables, showed that, for SLE patients, CD8 cells comprised the independent parameter associated with DN B cells; $R^2 = 0.277$, $p = 0.01$; b coefficient = 0.032; CI = 0.01–0.05; $p = 0.006$.

Therefore, CD8 subpopulations were included in a new multiple regression analysis, which revealed a model including CD8CD45RA-CD57- and CD8CD28+CD57- as the only independent factors related to DN B cells, with adjusted $R^2 = 0.464$, $p < 0.0001$.

The above model remained stable after adding the rest of the CD8 and CD4 subtypes, but was benefited by adding CD4CD31+ cells. Therefore, the model including CD8CD45RA-CD57-, CD8CD28+CD57-, and CD4CD31+ cells had an adjusted $R^2 = 0.534$, $p < 0.0001$.

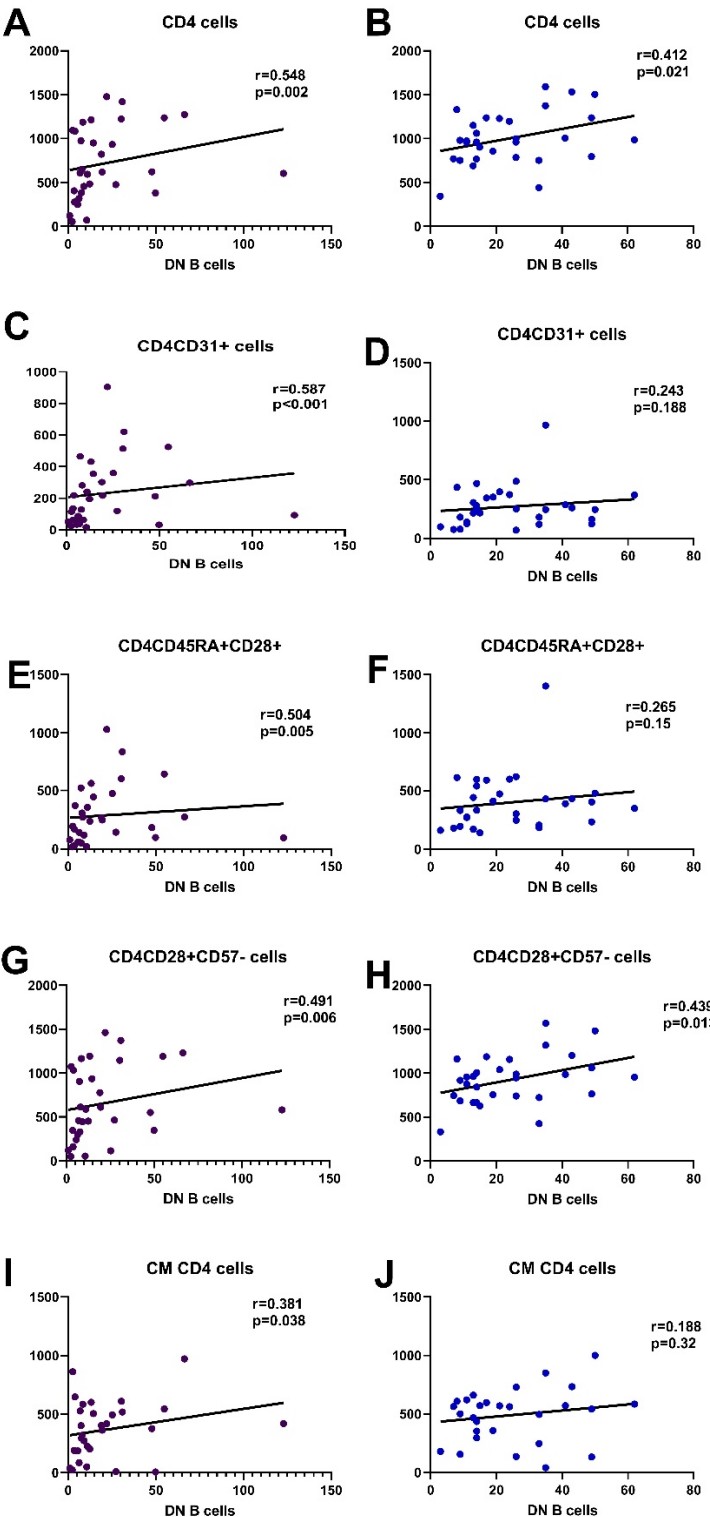

**Figure 5.** Most significant correlations of CD4 subpopulations with DN B cells in SLE patients and HC, CD4 ((**A**,**B**), respectively), CD4CD31 ((**C**,**D**), respectively), CD4CD45RA+CD28+ ((**E**,**F**), respectively), CD4CD28+CD57- ((**G**,**H**), respectively), and CM CD4 cells ((**I**,**J**), respectively).

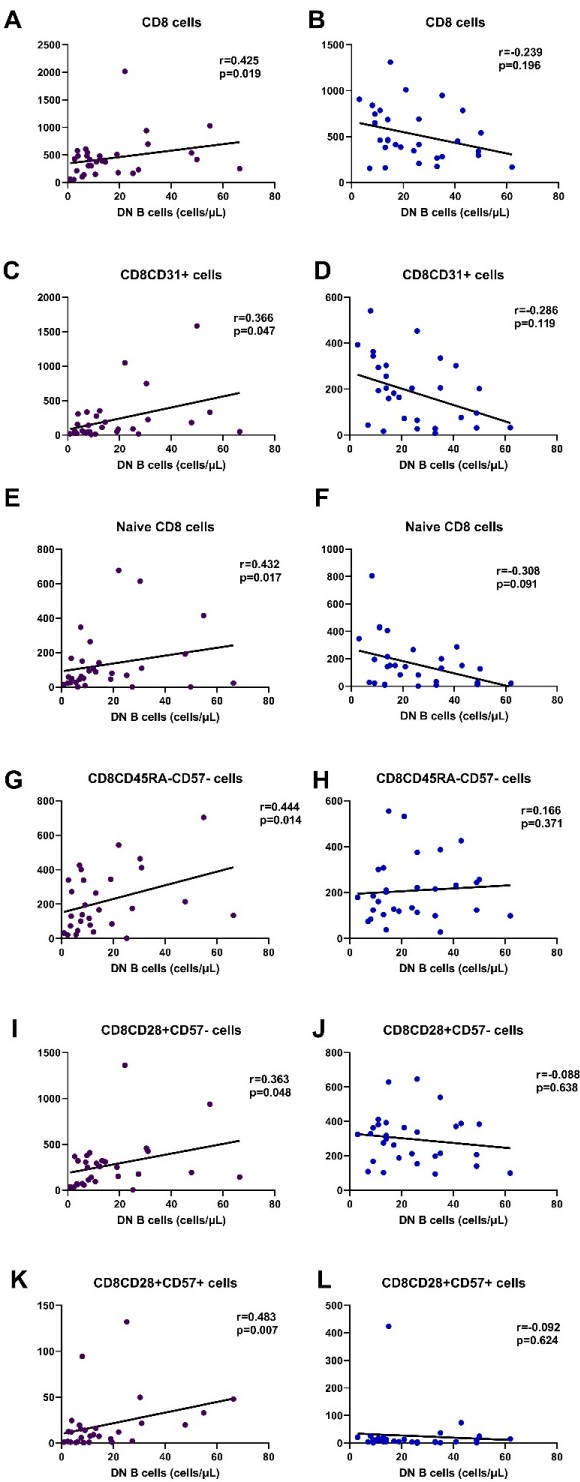

**Figure 6.** Most significant correlations of CD8 subpopulations with DN B cells in SLE patients (purple) and HC (blue), CD8 ((**A,B**), respectively), CD8CD31 ((**C,D**), respectively), naïve CD8 cells ((**E,F**), respectively), CD8CD45RA-CD57- ((**G,H**), respectively), CD8CD28+CD57- ((**I,J**), respectively), and CD8CD28-CD57- ((**K,L**), respectively).

## 4. Discussion

The present study was performed in an attempt to estimate a possible correlation between T cell immunity and DN B lymphocytes in patients with SLE. DN B cells consist of a certain subpopulation, which is significantly increased in the elderly but also in several chronic inflammatory diseases, including SLE [7,11,16–18]. Their origin, function, and

differential status is still unclear, as they carry characteristics of both naïve, switched memory, and immune-exhausted cells [6,7]. They may originate from an early termination of the GC reaction, from switched memory B cells, or from naïve resting B cells [6].

We have previously described the significant elevation, up to threefold, of DN B cells in SLE patients compared to healthy controls [11]. Here, we describe a close association of this subtype with certain, less differentiated CD4 and CD8 lymphocyte subpopulations, namely CD4CD31+, CD4CD45RA+CD28+, CD4CD45RA+CD57-, CD4CD45RA-CD57-, CD4CD28+CD57-, CD4CD28+CD57+, CD8CD31+, naïve CD8, CD8CD45RA+CD28+, CD8CD45RA-CD57-, CD8CD28+CD57-, CD8CD28+CD57+, and also CM CD4 cells. Very interestingly, almost all naïve or early differentiated T lymphocyte subsets studied showed a positive correlation with DN B cells, while no such association was evident for advanced differentiated T cells. Similar correlations were not evident in the healthy control population.

SLE, as well as other systemic autoimmune diseases, is associated with a significant increase in DN B lymphocytes, both DN1 and DN2. From those, DN2 B cells are most important, as they act as antigen-presenting cells and stimulate T cell response, differentiate into antibody and cytokine-secreting cells, and modulate immune response, sustained autoimmunity, and inflammation [16–19]. DN2 B lymphocytes share similar characteristics with naïve B lymphocytes, such as the absence of CXCR5, CD24, and CD38 surface molecules, which are common in the rest of peripheral B lymphocyte subsets, but also retain a unique phenotype, expressing CD69, HLA-DR, CD86, and the regulatory receptors CD32b and CD22, and lacking the lymph node homing receptor L-selectin (CD62L) [7,9,20].

Both DN2 and naïve B lymphocytes express a T-bet transcriptional network, with an upregulation of Toll-like receptor-7 (TLR7) and a simultaneous reduction in its negative regulators, such as TRAF5 [9]. As a result, DN2 and naïve B cells are hyper-responsive to TLR7. Stimulation with TLR7 induces phosphorylation of the ERK and MAPKp38 pathways, resulting in their continuing activation, the production of autoantibodies, and the generation of peripheral plasmablasts [21]. Previous studies have described the above changes in active SLE and active lupus nephritis [22], while others have demonstrated a higher rate of DN B cells in LN patients, with a positive correlation with the degree of proteinuria, certain immunological markers, such as C4 and ant-dsDNA titers, as well as clinical manifestations of the disease [23].

DN2 B cells account for a part of the so called "Age-Associated B cells" (ABCs), a heterogenous B cell population that, apart from elderly people, rises in patients with autoimmune diseases [24]. An interesting study presented another point of view and, except for the ABC subpopulation (also found expanded in SLE patients), examined the CD21hi subset. This study, despite the relatively small number of follow-up patients, exhibits outcomes regarding changes over induction therapy, namely the deterioration of ABCs and an increase in the CD21hi subset. These alterations appeared in accordance with complete and partial remission, suggesting a possible prognostic role as remission markers in LN [25].

Our patients displayed increased DN B lymphocytes while being in remission, indicating that despite clinical repression, the B cell compartment still demonstrates abnormal distribution. However, in the present study, we did not estimate the expression of CXRC5, the receptor which distinguishes DN1 (CXCR5+) from DN2 (CXCR5-) B lymphocytes. Previous studies have shown that DN2 cells are increased during disease flares and that their levels correlate with autoantibody production [9,22].

Although there are several studies confirming the predominance and expansion of DN B lymphocytes in SLE patients and a significant correlation with disease pathogenesis, there is scarcely any evidence regarding factors that may stimulate their production [26,27]. The correlation between DN B lymphocytes and recent thymic emigrant cells, as well as between almost all naïve CD4 and CD8 and central memory CD4 lymphocytes described in our study, is a completely novel finding which indicates, but cannot prove, causality. Previous studies have shown an increased expression of DN B lymphocytes in the cerebrovascular fluid of patients with multiple sclerosis, and revealed a T cell stimulatory and proinflammatory

function [28]. Similarly, in rheumatoid arthritis, DN B lymphocytes seemed to actively influence disease activity and were downregulated by anti-TNFa treatment [29]. However, in other studies, the significantly increased levels of DN B lymphocytes were characterized by a low expression of HLA-DR, CD40, and CD80 surface molecules, indicating their incapacity to act as antigen-presenting cells or interact with T lymphocytes [26]. In elderly patients, DN B cells seemed to behave more as senescent or exhausted cells than naïve B cells [30,31]. Senescent phenomena were not found to predominate in SLE [11,12], although they are prominent findings in chronic inflammatory diseases [32,33].

It is yet not specified whether the predominance of DN B cells in SLE, namely the DN2 subtype that originates from extrafollicular response, is an upfront outcome of a disturbed immunity response or a crucial pathogenetic agent [34]. In either case, interesting original studies on SLE murine models showed that the conditional deletion of T-bet-expressing B lymphocytes blocked the proper formation of the germinal center and the activation of B and T lymphocytes, accompanied by histological and clinical improvement [35]. These findings point to the crucial role of DN B cells in the pathogenesis of SLE and could account for a possible therapeutic target in the future.

Rituximab, a B cell-affecting agent, has also been studied regarding its effect on DN B cells, ABC-like B cells (CD11c+CD21- within the DN gate), and certain T cell subpopulations. In particular, ABC-like B cells were reduced from baseline within the first 2–4 months after treatment. On the other hand, while investigating T cell alteration, remarkable was the upregulation of effector memory and EMRA T cells and the downregulation of naïve T cells [36].

The activation of T lymphocytes and upregulation of DN B lymphocytes seem to occur simultaneously, and it remains unclear whether a common pathogenic mechanism orchestrates the whole reaction or DN B lymphocytes are themselves implicated in the stimulation of naïve, early differentiated, and memory peripheral T lymphocytes. Our findings of a close association between DN B cells and early differentiated T lymphocytes in patients with SLE raise the possibility that DN B lymphocytes are characterized by unique and important characteristics and play a major role in the alterations of adaptive immunity.

There are certain limitations in our study, including the absence of a group of SLE patients with active disease and the lack of differentiation of DN B lymphocytes into DN1 and DN2. However, according to the previous literature, increased levels of DN2 B cells have already been demonstrated in active SLE disease, and their pathogenetic role has also been proven in many aspects. While in remission, our patients were still on standard immunosuppressive treatment, including low doses of steroids in combination with low doses of MMF and/or cyclosporine. Previous studies have shown a reduction in the peripheral basophil population in adult and pediatric SLE and also demonstrated a reverse correlation with disease activity [37–39]. Although steroid treatment, even in low doses, could be implicated in the low numbers of peripheral basophils in these previous studies, this does not seem to be the case in our study because DN B lymphocytes are known to increase in active SLE, according to previous research, and we proved that they remain high despite steroid treatment and disease remission.

## 5. Conclusions

As we described in the present paper, peripheral double negative B lymphocytes were increased in our SLE patients in remission and had a significant correlation with naïve and early differentiated T lymphocyte subsets. These results indicate their important role in SLE and their involvement in T cell stimulation. As these results are described for the first time, we need to further assess changes in lymphocyte subpopulations and their interactions during disease evolution and regression following response to treatment.

**Supplementary Materials:** The following supporting information can be downloaded at: https://www.mdpi.com/article/10.3390/cimb45080421/s1. Table S1. Clinical and laboratory findings of patients with LN at time of diagnosis and time of evaluation; Figure S1. Gating strategy for CD4

and CD8 cells, and their subsets, based on the presence of CD45RA, CCR7, CD31, CD57 and CD28; Figure S2. Gating strategy for CD19+ cells, and their subsets based on the presence of IgD and CD27.

**Author Contributions:** Conceptualization, E.M. and M.S.; methodology, E.M.; software, D.I.B. and S.B.; validation, A.X., G.L. and M.C.; formal analysis, D.I.B.; investigation, M.C. and S.S.; resources, A.I. and M.I.; data curation, D.I.B.; writing—original draft preparation, E.M.; writing—review and editing, A.F. and M.S.; supervision, A.F. and M.S.; funding acquisition, M.S. All authors have read and agreed to the published version of the manuscript.

**Funding:** This research received no external funding.

**Institutional Review Board Statement:** The Institutional Review Board of the. The study was conducted in accordance with the Declaration of Helsinki, and approved by the Institutional Review Board of the Medical School of Aristotle University of Thessaloniki, Greece. Protocol code 254, date of approval 15/3/2020.

**Informed Consent Statement:** Informed consent was obtained from all subjects involved in the study.

**Data Availability Statement:** All data are available upon request from the authors.

**Conflicts of Interest:** The authors declare no conflict of interest.

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
