# Peer review of "Increase in Double Negative B Lymphocytes in Patients with Systemic Lupus Erythematosus in Remission and Their Correlation with Early Differentiated T Lymphocyte Subpopulations"

_cimb, doi:10.3390/cimb45080421_

Round 1

Reviewer 1 Report

Comments and Suggestions for Authors

The authors compared a relatively small group of patients with SLE in remission (30 pts) with similar number of healthy controls and were able to demonstrate that SLE patients have higher level of double negative B cells correlating with higher number of early activated T cells suggesting the role of double negative B cells in T cell activation.

Interesting data with potential clinical importance.  As the topic is not easy for (not only) and newcomer in the field I would appreciate a figure explaining the development of different subpopulations of B cells and T cells with the emphasis on difference in SLE and healthy subjects

Minor comments:

1.       Are there any data on DN B cells in non-treated patients with active SLE and their response to induction treatment with CPH, MMF, or monoclonals?

2.       Was there any difference between SLE patients based on the degree of remission (complete versus partial)? May the normalization of the number (proportion) of DN B cells be one of the markers of complete remission?

3.       Was there any difference in DN B cells  in respect to organ involvement (kidney)?

4.       I would appreciate any dicussion on the putative influence of the treatment on the number of the different subtypes of B cells and T cells in SLE. One would expect tha even hydroxycholoquine only should have an important impact mediated by the regulation of TLRs

5.       May DN negative B cells be a therapeutic target in SLE?

Author Response

Thank you very much for your effort to improve the manuscript, we have seen all comments and suggestions, and we revised the paper. 

Here are the answers to your comments

 Minor comments:

  1. 1.     Are there any data on DN B cells in non-treated patients with active SLE and their response to induction treatment with CPH, MMF, or monoclonals?

Thank you so much for your comment. Our search did not reveal any data regarding the effect of steroids or MMF on DN B cells, but there is a paper on the role of Rituximab, which is described in the paragraph:

Rituximab, a B cell affecting agent …………… downregulation of Naïve T cells[36]. 

  1. Was there any difference between SLE patients based on the degree of remission (complete versus partial)?

Thank you so much for this very important comment. We did not perform this analysis initially, but we did it now. After your suggestion, we evaluated differences in lymphocyte subpopulations between patients with complete or partial remission of LN. We did not find any significant difference, however, we must say that our patient population was small, this kind of subgroup analysis should involve a greater number of patients.

May the normalization of the number (proportion) of DN B cells be one of the markers of complete remission?

This is a very interesting point. Although DN B cells seem to play an important and unique role in SLE, we describe here that their levels are still increased even at remission, so they cannot be used as markers of outcome. However, the DN2 B cells seem to be more important, and further studies should focus on these B cell subtype.

We have added a paragraph in the discussion about the possible reduction of DN B cells following treatment, and their potential use as biomarkers DN2 B cells accounts as part of the so called “Age-Associated B cells” (ABCs)…………………… there possible prognostic role as remission markers in LN [25]

  1. Was there any difference in DN B cells in respect to organ involvement (kidney)?

Thank you very much for this valuable comment. We performed a subgroup analysis between patients with and without LN. We decided not to present the results for the following reasons

  1. There was no difference in B lymphocytes and their subpopulations, including Double Negative B cells, the population under investigation,
  2. Regarding the T lymphocyte compartment, there was a significant increase of advanced differentiated CD4 and CD8 cells (CD4-EMRA and CD8-EMRA) and exhausted CD4 cells. However, as these cell types had no correlation with DN B cells, we did not proceed to further investigation

iii. The number of SLE patients without LN was very small, and we cannot account to draw reliable conclusions

We added some discussion about the correlation of DN B cells with immune and clinical parameters. Previous studies have described…………….. as well as with clinical manifestations of the disease[23].

  1. 4.      I would appreciate any discussion on the putative influence of the treatment on the number of the different subtypes of B cells and T cells in SLE. One would expect that even hydroxycholoquine only should have an important impact mediated by the regulation of TLRs

        Thank you very much for your comment, we have added few paragraphs in the discussion with possible therapeutic targets aiming to T or B lymphocytes:

DN2 B cells accounts as part of the so called…………. role as remission markers in LN [25].

It is yet not specified ………. possible therapeutic target in the future

Rituximab, a B  cell …………… downregulation of Naïve T cells[36]. 

  1. May DN negative B cells be a therapeutic target in SLE?

You are absolutely right, DN B cells could be a therapeutic target in SLE, however, although there is evidence of the beneficial effect of T-bet expressing B lymphocytes elimination, more studies are needed to confirm these findings. Also, as different receptors are expressed on DN B cells, the research should focus on the molecules mostly affected by treatment.

Reviewer 2 Report

Comments and Suggestions for Authors

In the present study authors compared double negative ( CD19+IgD-CD27-  B cell count) in patients with systemic lupus in comparison to healthy control subjects and their relationship with naive, memory and advanced differentiated/senescent T cells.  Both B and T lymphocytes were significantly reduced in SLE patients. However, the percentage of DN B cells were increased, compared  to controls.   Distribution of CD4 and CD8 lymphocytes demonstrated 22 a shift to advanced differentiated subsets.  The population of DN B cells had significant positive 23 correlation with most of early differentiated T lymphocytes.   CD4CD31+, CD8CD45RA-CD57- and D8CD28+CD57- cells are independently associated with DN B cells in multivariate regression analysis.

I have the following comments regarding this paper:

1)     Was the required number of patients calculated before the study?

2)     Line 154: “continues” should be corrected to: “continuous”

3)     Line 159: why some data are presented as the full range (min-max) whereas others as IQ range?

4)     Was the fraction of males and females different or not between groups according to statistical analysis?

5)     Did the authors perform subgroup analysis in patients with and without LN?

Author Response

Thank you very much for your help to improve the paper

We have made the revisions in the manuscript, and we resubmit it

Here are the answers to your comments

1)     Was the required number of patients calculated before the study?

Thank you very much for your comment. We used the EpiTools sample size calculator (http://epitools.ausvet.com.au) to calculate the required number of patients. The expected percentage of SLE patients who present certain lymphocyte alterations was based on previous studies, however, as we evaluated a whole cohort of lymphocyte markers, there are not enough data for most of the subpopulations included in this study. Based on the large number of surface molecules studied, with an 1/1 patient/healthy control ratio, 95% confidence interval and 80% study power, the calculated result were 25 patients and 25 healthy controls. We included 30 patients and 31 healthy controls.

2)     Line 154: “continues” should be corrected to: “continuous”

Thank you for this correction, we changed it

3)     Line 159: why some data are presented as the full range (min-max) whereas others as IQ range?

Thank you for this comment, that was done by mistake, we have changed it by presenting all data as Mean±SD or Median(range)

4)     Was the fraction of males and females different or not between groups according to statistical analysis?

Yes, there was a significant preponderance of females in SLE patients, and this is described in the results. Females predominated in the SLE patients group, however, all parameters tested did not show any difference between males and females, neither in the patients, nor in the HC group.

However, we had no difference in any of lymphocyte subtypes between male and female neither in patients nor in healthy control group 

5)     Did the authors perform subgroup analysis in patients with and without LN?

Thank you very much for this valuable comment. We performed a subgroup analysis between patients with and without LN. We decided not to present the results for the following reasons

  1. There was no difference in B lymphocytes and their subpopulations, including Double Negative B cells, the population under investigation,
  2. Regarding the T lymphocyte compartment, there was a significant increase of advanced differentiated CD4 and CD8 cells (CD4-EMRA and CD8-EMRA) and exhausted CD4 cells. However, as these cell types had no correlation with DN B cells, we did not proceed to further investigation

iii. The number of SLE patients without LN was very small, and we cannot account to draw reliable conclusions

Reviewer 3 Report

Comments and Suggestions for Authors

INTRODUCTION

- Overall, it is well balanced. However, most concepts from lines 62 may be more suitable to the discussion rather than the introduction, which aims to provide a general background only.

- The final paragraph should schematically state the general study objectives

METHODS

- I think this section needs some rearrangements.

- The first section should be study design and population.

- The study period is unclear.

- It is not clear why only SLE patients in remission were included (no one at first diagnosis). In this regard, in light of this limitation, the title should be revised. 

- Healthy controls are not defined at all in terms of clinical background and recruitment.

- Ethical statement subsection should be added. In addition to the IRB approval number and date, the type of informed consent should be clarified.

- Were all SLE patients consecutive? If not, also clarify why some were excluded or not recruited.

- One section on flow cytometry would be enough. You can merge them.

- The resolution of supplementary figures should be improved.

- The number of lupus nephritis patients should be included in the results.

- Moreover, the therapy should be clearly reported in the results (“SLE should be stable with no flare up for at least 2 years, and their immunosuppression, at time of evaluation, could include prednisolone, hydroxychloroquine, calcineurin inhibitors and azathioprine or mycophenolate mofetil”).

RESULTS

- The full demographic and clinical characteristics of SLE patients should be presented.

- A different table showing all the more relevant laboratory parameters, including the hematological counts and inflammatory indexes should be drafted.

- Overall, more detailed clinical and laboratory data should be given.

- I think that Figure 1 should be changed with a more appropriate type of graphical representation, perhaps just columns. The current representations may be misunderstood as it was a timeline.

- Similar concerns as regards figure 3.

- As mentioned above, detailed therapy should be provided for the study population.

- Table 6 could be better represented with a figure.

DISCUSSION

- To be assessed after methodological clarifications and results completion.

- However, the point about the therapy is essential. For instance, adult SLE patients were also reported to have alterations of basophils homeostasis (see: Clin. Rheumatol. 2018;37:459–465. doi: 10.1007/s10067-017-3858-4); this finding was confirmed in SLE children as well, but the authors raised the issues of the influence of steroid therapy on this result (refer to: Diagnostics (Basel). 2022 Jul 12;12(7):1701. doi: 10.3390/diagnostics12071701), which seems to be the case of the present study as well. These aspects and limitations should be carefully discussed.

- By the way, a clear discussion about the study limitations is missing.

- There is no clear conclusion section.

Comments on the Quality of English Language

See comments above

Author Response

Thank you very much for your valuable comments, and for your effort to improve the manuscript. We have revised it according to your suggestions, and we submit it again

Here are the answers to your comments

INTRODUCTION

- Overall, it is well balanced. However, most concepts from lines 62 may be more suitable to the discussion rather than the introduction, which aims to provide a general background only.

Thank you very much for this valuable comment, we have made changes in the introduction section and focused on a brief presentation of DN B cells

- The final paragraph should schematically state the general study objectives

Thank you very much for this suggestion, we changed the final paragraph of the introduction section

METHODS

- I think this section needs some rearrangements.

- The first section should be study design and population.

Thank you very much for this point, we have changed it

- The study period is unclear.

Thank you for this interesting comment, this is a cross sectional study, we describe this in section 2.1

- It is not clear why only SLE patients in remission were included (no one at first diagnosis).

Thank you very much for this comment, we decided to include SLE patients in remission for two reasons 1) because the predominance of DN B cells in active SLE is well described previously, and there is a question if these cells are reduced after immunosuppression treatment, during remission, and 2) we could have an extra group of active SLE patients, but this would need much more time. For these reasons, based on previous research, we assume that DN B cells are proved to be increased in active SLE and we decided to assess their levels in patients at remission.

In fact, we have already started a similar study, evaluating this population at time of first diagnosis, or at time of relapse and reassess after remission, the results are expected to give more accurate information

In this regard, in light of this limitation, the title should be revised. 

Thank you very much, you are absolutely right, we have changed the manuscript title

- Healthy controls are not defined at all in terms of clinical background and recruitment.

Thank you very much for your comment, we have added a paragraph with the inclusion/exclusion criteria for the HC group.

Inclusion-Exclusion criteria for HC group: Volunteers eligible to participate to HC group should be Caucasians, 18-67 years old, with no comorbidities, as described above, normal renal function, they should not take any medication, and caution was taken to be of similar sex with patients.

- Ethical statement subsection should be added. In addition to the IRB approval number and date, the type of informed consent should be clarified.

Thank you for your comment, we have added the IRB approval number, protocol code 748/21, 3 October 2021

- Were all SLE patients consecutive? If not, also clarify why some were excluded or not recruited.

Thank you so much for this suggestion. We evaluated 37 patients consecutively, from whome we enrolled 30, the reasons for excluding 7/37 are explained, we have added a paragraph in the beginning of the results section.

Thirty seven patients were consecutively evaluated initially, from whom 7 were excluded, 3/7 because of a recent (<2 years) relapse, 2/7 because they had rituximab in the past, and 1/7 had a history of melanoma and 1/7 ad developed diabetes mellitus.

- One section on flow cytometry would be enough. You can merge them.

Thank you, we have changed this

- The resolution of supplementary figures should be improved.

Thank you, we have changed figures from JPEG to TIF, and improved the resolution to 1200

- The number of lupus nephritis patients should be included in the results.

Thank you, we have added this information, in the results section Twenty two patients had LN, diagnosed with renal biopsy, performed not more than 5 years prior to evaluation, and also, the Supplement Table 1.

- Moreover, the therapy should be clearly reported in the results (“SLE should be stable with no flare up for at least 2 years, and their immunosuppression, at time of evaluation, could include prednisolone, hydroxychloroquine, calcineurin inhibitors and azathioprine or mycophenolate mofetil”)

Thank you very much for this comment, we give detail of the treatment and dosages in the results section.

Their treatment included hydroxychloroquine in 30/30 (100%), prednisolone (5-7.5mg/d) in 24/30 (80%), MMF (500-1000mg/d) in 18/30 (60%) and CNIs (1.5mg/Kg/d) in 9/30 (30%) or combination of MMF+CNIs in 5/30 (16.7%). Time since patients were on the above regimes were 84(45-125)months for hydroxychloroquin, 78(45-92)months for steroids, 53(37-65)months for MMF and 42(28-56)months for CNIs.

RESULTS

- The full demographic and clinical characteristics of SLE patients should be presented.

Thank you very much for this comment, we added a table (Supplement table 1 with demographic and clinical characteristics of the SLE patients

- A different table showing all the more relevant laboratory parameters, including the hematological counts and inflammatory indexes should be drafted.

Thank you very much for this suggestion, we present all laboratory data at table 1

- Overall, more detailed clinical and laboratory data should be given.

Thank you, that was a very important comment, we present all demographic , clinical and laboratory data of the patients at tables Suppl Table 1 and Table 1

- I think that Figure 1 should be changed with a more appropriate type of graphical representation, perhaps just columns. The current representations may be misunderstood as it was a timeline.

- Similar concerns as regards figure 3.

Thank you so much for this suggestion, we have changed all these figures to boxplots

- As mentioned above, detailed therapy should be provided for the study population.

Thank you, we have changed this

- Table 6 could be better represented with a figure.

Thank you for this comment, you are absolutely right, the figure could demonstrate the correlation clearly and it is easier to read. However, the large number of correlations (fourteen for SLE patients: 8 significant and 6 non-significant and for 14 for HC, 6 significant and 8 non-significant) will make it very difficult to demonstrate. For that reason, we have changed table 5, which shows correlations between DN B cells with B lymphocytes and their subpopulations, and left table 6, which became table 5. However, if you believe we have to change the table, we can do it.

DISCUSSION

- To be assessed after methodological clarifications and results completion.

- However, the point about the therapy is essential. For instance, adult SLE patients were also reported to have alterations of basophils homeostasis (see: Clin. Rheumatol. 2018;37:459–465. doi: 10.1007/s10067-017-3858-4); this finding was confirmed in SLE children as well, but the authors raised the issues of the influence of steroid therapy on this result (refer to: Diagnostics (Basel). 2022 Jul 12;12(7):1701. doi: 10.3390/diagnostics12071701), which seems to be the case of the present study as well. These aspects and limitations should be carefully discussed.

Thank you very much for this comment, we have added this in the discussion, paragraph before the last one, However, according to previous literature …………. steroid treatment and disease remission.

- By the way, a clear discussion about the study limitations is missing.

Thank you, we have added this

There are certain limitations in our study, including the absence of a group of SLE patients with active disease and the lack of differentiation of DN B lymphocytes to DN1 and DN2.

- There is no clear conclusion section.

Than you, we added a conclusion section

In conclusion, the increase levels of peripheral double negative B lymphocytes in SLE patients at remission, and their significant correlation with naïve and early differentiated T lymphocyte subsets, indicate their important role in SLE, and their involvement in T cell stimulation. As these results are described for the first time, we need to further assess changes in lymphocyte subpopulations and their interactions, during disease evolution and regression, following response to treatment.   

Round 2

Reviewer 2 Report

Comments and Suggestions for Authors

The manuscript has been revised according to the reviewers' comments.

Author Response

Dear Reviewer

Thank you for your help to improve our paper

Reviewer 3 Report

Comments and Suggestions for Authors

The authors addressed most of my previous comments, but I would like to ask for some further specifications, as follows.

- The study period should be indicated.

- It is still unclear the procedure of enrollment of the HC. Where were they recruited? 

- I would suggest creating a separate (from the discussion) conclusion section 

- Table 5: is it possible to represent the correlation plot as a figure with appropriate colors and values, instead of as a table. Which would be clearer for the readership, in my opinion.

Comments on the Quality of English Language

See above

Author Response

Thank you for your comments and effort to impove our manuscript

We made all suggested corrections, and we resubmit our paper

Here are the asnwers to your comments

- The study period should be indicated.

Thank you very much for your comment; we recruited patients and HC during a 6 months time

All patients were followed up in the SLE outpatients clinic, and recruited in a 6 months time.

- It is still unclear the procedure of enrollment of the HC. Where were they recruited? 

Thank you for your comment, we describe that healthy controls were recruited from medical, paramedical personnel and patients partners, not relatives.

HC were either patients’ partners or medical or paramedical personnel.

- I would suggest creating a separate (from the discussion) conclusion section

Thank you for this suggestion, you are absolutely right, we created a separate section with the conclusion. 

- Table 5: is it possible to represent the correlation plot as a figure with appropriate colors and values, instead of as a table. Which would be clearer for the readership, in my opinion.

Thank you very much, you were right, we replaced table 5 to two figures 5 and 6, for CD4 and CD8 subpopulations respectively, and we moved table 5 to the supplement material, as supplement table 2